The old and the new plankton: ecological replacement of associations of mollusc plankton and giant filter feeders after the Cretaceous?

Tajika Amane 1
Nützel Alexander 2
Klug Christian chklug@pim.uzh.ch 1
1 Paläontologisches Institut und Museum, Universität Zürich , Zürich , Switzerland
2 SNSB-Bayerische Staatssammlung für Paläontologie und Geologie, Department of Earth and Environmental Sciences, Palaeontology & Geobiology, GeoBio-Center LMU , München , Germany
De Baets Kenneth
Electronic publication date: 2018 Jan 9
Publication date: 2018
Volume: 6
Electronic Location ID: e4219
Received 2017 Sep 25; Accepted 2017 Dec 12
Copyright: ©2018 Tajika et al.
Copyright year: 2018
Copyright holder: Tajika et al.
License: This is an open access article distributed under the terms of the Creative Commons Attribution License, which permits unrestricted use, distribution, reproduction and adaptation in any medium and for any purpose provided that it is properly attributed. For attribution, the original author(s), title, publication source (PeerJ) and either DOI or URL of the article must be cited.
License URL: https://creativecommons.org/licenses/by/4.0/

Keywords: Belemnitida, Ammonoidea, Cretaceous, Fecundity, Palaeogene, Filter feeders, Holoplanktonic gastropoda, Pachycormiformes, Mass extinctions

Funding: Swiss National Science Foundation SNF 200021_149119 Deutsche Forschungsgemeinschaft NU 96/13-1 This study is supported by the Swiss National Science Foundation SNF (project number 200021_149119). This study was also supported by the Deutsche Forschungsgemeinschaft NU 96/13-1. The funders had no role in study design, data collection and analysis, decision to publish, or preparation of the manuscript the Deutsche Forschungsgemeinschaft NU 96/13-1.

==============================
Owing to their great diversity and abundance, ammonites and belemnites represented key elements in Mesozoic food webs. Because of their extreme ontogenetic size increase by up to three orders of magnitude, their position in the food webs likely changed during ontogeny. Here, we reconstruct the number of eggs laid by large adult females of these cephalopods and discuss developmental shifts in their ecologic roles. Based on similarities in conch morphology, size, habitat and abundance, we suggest that similar niches occupied in the Cretaceous by juvenile ammonites and belemnites were vacated during the extinction and later partially filled by holoplanktonic gastropods. As primary consumers, these extinct cephalopod groups were important constituents of the plankton and a principal food source for planktivorous organisms. As victims or, respectively, profiteers of this case of ecological replacement, filter feeding chondrichthyans and cetaceans likely filled the niches formerly occupied by large pachycormid fishes during the Jurassic and Cretaceous.

Introduction

The fate of individual groups of marine organisms at mass extinctions is of considerable interest (e.g., Jablonski & Raup, 1994; Jablonski, 2008). By contrast, the disappearance of entire communities or ecological associations or food webs or important parts of any of these structures from the geologic past still requires a lot of palaeontological research (Hautmann, 2014; Hofmann et al., 2014; Roopnarine & Angielczyk, 2015). Extinctions of entire communities or ecosystems are most conspicuous during the great mass extinctions (e.g., Foster & Twitchett, 2014), when usually vast new ecospace was freed and thereby, new ecological niches could form during recovery periods.

Although it is not the most severe of the Big Five, the end-Cretaceous mass extinction is likely the most famous among those with the greatest severity (McGhee et al., 2013). This fame roots in the facts that popular groups of organisms such as dinosaurs (Sloan et al., 1986; Archibald & Fastovsky, 2004) and ammonites (Goolaerts, 2010; Kennedy, 1993; Landman et al., 2015) were erased by the consequences of an impact in Mexico and/or flood basalt-eruptions in India (Keller et al., 2009; Miller et al., 2010; Schulte et al., 2010; Tobin et al., 2012).

Marine communities were heavily affected as reflected in a significant reduction of diversity or a partial or even total disappearance of major groups such as ammonoids and belemnites (Doyle, 1992; Marshall & Ward, 1996; Iba et al., 2011; Olivero, 2012; Landman et al., 2014) as well as planktonic foraminifers (Alvarez et al., 1980; Smit, 1982) and bivalves (Jablonski & Raup, 1994). Before this extinction, ammonoids were both highly diverse and had evolved a great disparity in the course of the Cretaceous (Ward & Signor, 1983; Ward, 1996); some of the most bizarre forms (heteromorphs) such as Nipponites, Diplomoceras and Didymoceras appeared, some of which reached impressive sizes (e.g., Diplomoceras, Emericiceras). Furthermore, members of the family Puzosiidae, which comprises the largest ammonoids of all times, also lived during Cretaceous times (Landois, 1895; Olivero & Zinsmeister, 1989; Kennedy & Kaplan, 1995). Puzosiids are not only gigantic but they also abundantly occurred worldwide. In addition to this family, members of other ammonite families reached sizes of around one meter in the Late Cretaceous as well.

The sometimes extreme variation in conch size and morphological disparity makes it likely that their modes of life and habitats differed as well. This is reflected in hatchling conch morphology, which may produce shapes unknown from adult ammonoid conchs (Klug, De Baets & Korn, 2016). A further line of reasoning supporting ontogenetic changes in ecology is the change in hydrodynamic properties due to size changes (Jacobs, 1992; Jacobs & Chamberlain, 1996; Naglik et al., 2015).

The great abundance, wide geographical distribution, extreme diversity, middle to giant size in combination with the likely high fecundity of ammonites raises a number of questions: (i) what respective roles did juvenile and adult ammonites and belemnites play in Cretaceous marine food webs? (ii) The adults were probably planktotrophic consumers, but what was the role of their minute offspring? (iii) What groups filled the ecospace occupied by ammonites, belemnites and their predators after the Cretaceous? (iv) How did this ecological replacement occur?

Figure 1 Adult ammonites (A–C), juvenile ammonites (D, E), and an embryonic belemnite (F) compared to fossil conchs of Thecosomata from the Eocene of India (G–J). 0.1 mm-scale bar applies to figures D to J.

Photo in (A) courtesy C Steinweg, L Schöllmann and J-O Kriegs (all Mnster); (D and E) redrawn after Tanabe, Kulicki & Landman (2008); (F) from Bandel et al. (1984); (G to J) redrawn after Lokho & Kumar (2008). (A) Parapuzosia seppenradensis, Campanian, Seppenrade. (B) C. Pachydesmoceras sp., Campanian, Hokkaido, diameter 1.3 m, D. Aiba (Mikasa) for scale. Note the symmetrical bulges in the posterior body chamber in C. (D) juvenile conch of Scaphites whitfieldi, AMNH 44833, Turonian, USA. (E) embryonic conch of Aconeceras cf. trautscholdi, UMUT MM 29439–4, Aptian, Russia. (F) embryonic conch of Hibolithes sp., GPIT Ce 1599, Callovian, Lithuania. (G) to (J), Upper Disang Formation, Phek District, Nagaland. (G, H) Limacinidae spp. (I, J) Creseidae spp.

Methods

We estimated the fecundity of large Cretaceous ammonites such as Parapuzosia seppenradensis (Fig. 1) using the following facts, assumptions and measurements, which probably applied to all members of the Ammonoidea. (i) we know that the major part of egg-development happened in the body chamber (De Baets, Landman & Tanabe, 2015; Mironenko & Rogov, 2016); (ii) there is good evidence that the ammonitella represents the embryonic part of the conch (De Baets, Landman & Tanabe, 2015); (iii) we suggest that egg-size only slightly exceeded ammonitella-size because of their dense packing in fossils with embryos preserved in the body chamber (De Baets, Landman & Tanabe, 2015; Mironenko & Rogov, 2016); and (iv) we followed the proportion of 8% of the soft body volume being occupied by the gonads according to the proportions known from Recent Nautilus (Tanabe & Tsukahara, 1987; Korn & Klug, 2007; De Baets, Landman & Tanabe, 2015). As far as (iv) is concerned, there is some uncertainty because the proportions of the ovaries are poorly known from ammonoids due to the extremely rare and fragmentary preservation of soft parts (Mironenko & Rogov, 2016; Lehmann, 1981; Lehmann, 1985; Klug & Lehmann, 2015; Klug, Riegraf & Lehmann, 2012). When regarding the specimens figured by Mironenko & Rogov (2016), one tends to assume that the gonads filled a much larger portion of the body chamber. This hypothesis finds further support in symmetric bulges in the posterior body chamber in mature Pachydesmoceras (Fig. 1) and scaphitid conchs (Kennedy, 1989). These bulges may have offered space for the growing ovaries. Owing to these materials and morphological adult modifications of ammonoid conchs, we calculated alternative maximum egg-numbers using a body chamber volume proportion occupied by gonads of 30%.

The largest specimen of the largest ammonite species Parapuzosia seppenradensis is incomplete (Landois, 1895; Kennedy & Kaplan, 1995). This specimen measures 1,740 mm (Fig. 1). We estimated the adult body chamber volume and the surface area of the terminal aperture assuming a body chamber length of about 180 degrees because of shell traces of the missing conch part along the umbilical seam. Accordingly, the maximum diameter dm can be reconstructed to have reached 2,200 mm with a whorl height wh of about 800 mm and a whorl width ww of about 500 mm. The radiuses would then measure 1,250 mm (r1) at the terminal aperture and 950 mm (r2) on the opposite side. Using the wh and ww values, we reconstructed a whorl cross section in CorelDraw and measured the area; accordingly, the cross section area K amounts to almost 320,000 mm2.

As shown by De Baets et al. (2012), derived ammonoids (i.e., with fully coiled embryonic conchs) likely had a high fecundity. This is corroborated by the great differences between embryo size and adult conch size. For example, in the largest specimen of Parapuzosia seppenradense from the Late Cretaceous of Germany, the embryonic conch measured about one millimeter in diameter at hatching, while the adult conch exceeded two meters in diameter (Kennedy & Kaplan, 1995; De Baets et al., 2012; De Baets, Landman & Tanabe, 2015; Korn & Klug, 2007; Landman, Tanabe & Shigeta, 1996; Tanabe, Kulicki & Landman, 2008). This implies a factor of at least 2,000 in diameter increase between embryos and adult macroconchs. Embryonic conch size (ammonitella size) is well documented for most ammonoid clades (De Baets, Landman & Tanabe, 2015). In Cretaceous ammonoids, ammonitella size ranges between 0.5 and 1.5 mm with the average being smaller than 1 mm (De Baets, Landman & Tanabe, 2015).

In order to estimate the absolute gonad volume, we determined the body chamber volume VBC, which can be achieved by applying an equation introduced by Raup & Chamberlain (1967) and also used by De Baets et al. (2012): (1) VBC=2∕3⋅π⋅K⋅Ra∕lnW⋅1−W−3θ∕2π

with K—area of the last aperture, Ra—distance coiling axis to center of mass (estimated 200 mm based on comparisons with species with similar conch shape: Tajika et al., 2015; Naglik, Rikhtegar & Klug, 2016), θ—angular length of the body chamber in radians (equals π here, because the body chamber is about 180° long), the whorl expansion rate for this particular body chamber length (2) W=r1∕r22π∕θ

with r1—radius at maximum conch diameter and r2—radius at conch diameter 180° behind the aperture.

In order to compare the embryonic conch size of Cretaceous ammonites to early Thecosomata, we collected data from various publications (for values and references, see Tables 1, 2 and 3. We plotted these values in box plots using Excel.

As a further approach to estimate ammmonoid fecundity, we gathered published data on the numbers of eggs per adult female and list those in Table 4 (references of the data sources are given there). Using Excel, we produced a loglog-biplot depicting the relationship between the estimated number of eggs and the mature conch size in various ammonoids of Devonian to Cretaceous age. Unsurprisingly, there is an exponential relationship between the two parameters.

Results

Estimating ammonoid fecundity

Applying the data and calculations listed in the method section to the lectotype of Parapuzosia seppenradense, we obtain a whorl expansion rate W of 1.73 and then an according body chamber volume VBC of 137,075,470 mm3. Depending on the proportion of the gonads (between 8 and 30%; see discussion in ‘Methods’), we obtain gonad volumes varying between about 10,000,000 mm3 and 40,000,000 mm3. Assuming an egg-volume of 1 mm3, we obtain numbers of 10,000,000 to 40,000,000 eggs per adult female Parapuzosia seppenradensis (see also Fig. 2) if they were semelparous. If we assume iteroparity, these numbers increase by the factor of the number of reproductive cycles. Also, if we assume that the eggs and embryos continued to grow after they were laid (see discussion in Walton, Korn & Klug, 2010), ammonoid fecundity would further increase, but evidence for this is missing in ammonoids (Mironenko & Rogov, 2016). For an adult female of half the diameter, we would still obtain egg-numbers of between 3,000,000 (8% gonad volume) and 10,000,000 eggs (30% gonad volume) at semelparity. Puzosiids and other large Cretaceous ammonoids in the size range between 500 and 1,000 mm are quite common worldwide (e.g., Pachydesmoceras).

In Fig. 2, we depict the relationship between the number of eggs per female adult ammonoid using mostly published data and the results presented here. The loglog-biplot shows an exponential relationship between these two parameters, which is not surprising taking into account how the estimates were achieved (De Baets et al., 2012). It also shows some variation scattering around the exponential trend line, which likely roots in the variation of the ratio of embryonic versus adult conch size (e.g., Gyroceratites and Sinzovia have similar adult sizes, but the embryonic conch is 50% larger in Gyroceratites; De Baets et al., 2012; De Baets, Landman & Tanabe, 2015; Mironenko & Rogov, 2016).

Table 1 Measurements of embryonic conchs of Cretaceous ammonites.

Data from De Baets, Landman & Tanabe (2015) and Laptikhovsky, Nikolaeva & Rogov (2017).

Species	Stage	AD mean	
Acanthoplites sp.	Albian	0.75	
Aconeceras (Sanmartinoceras) sp.	Aptian	0.84	
Aconeceras trautscholdi	Aptian	0.63	
Aconeceras trautscholdi	Aptian	0.77	
Anagaudryceras limatum	Coniacian	1.28	
Anagaudryceras matsumotoi	Maastrichtian	1.32	
Anagaudryceras nanum	Campanian	1.26	
Anagaudryceras tetragonum	Maastrichtian	1.26	
Anagaudryceras yokoyamai	Santonian	1.41	
Baculites sp.	Santonian	0.78	
Beudanticeras beudanti	Albian	1.06	
Beudanticeras laevigatum	Albian	0.88	
Boreophylloceras densicostatum	Berriasian	2.37	
Boreophylloceras praeinfundibulum	Berriasian	2.9	
Calliphylloceras subalpinum	Albian	0.8	
Calliphylloceras velledae	Aptian	0.84	
Calycoceras orientale	Cenomanian	0.93	
Canadoceras kossmati	Campanian	0.89	
Canadoceras mystricum	Campanian	1	
Clioscaphites vermiformis	Santonian	0.71	
Collignoniceras woollgari	Turonian	0.82	
Colombiceras sp.	Aptian	0.62	
Damesites ainuanus	Turonian	0.7	
Damesites damesi	Santonian	0.89	
Damesites latidorsatus	Santonian	0.85	
Damesites semicostatus	Santonian	0.84	
Damesites sugata	Santonian	0.83	
Deshayesites deshayesi	Albian	1	
Desmoceras dawsoni	Albian	1.52	
Desmoceras ezoanum	Cenomanian	1.22	
Desmoceras japonicum	Turonian	0.97	
Desmoceras kossmati	Cenomanian	0.85	
Desmoceras kossmati	Cenomanian	0.9	
Desmoceras poronaicum	Albian	0.84	
Desmophyllites diphylloides	Santonian	0.86	
Desmophyllites diphylloides	Santonian	0.81	
Desmophyllites sp.	Santonian	0.84	
Desmophyllites sp.	Santonian	0.89	
Diadochoceras nodosocostatiforme	Aptian	0.77	
Diadochoceras sinuosocostatus	Aptian	0.74	
Diadochoceras sp.	Aptian	0.75	
Discoscaphites conradi	Maastrichtian	0.76	
Discoscaphites gulosus	Maastrichtian	0.72	
Discoscaphites rossi	Maastrichtian	0.69	
Eogaudryceras (Eotetragonites) aurarium	Albian	0.93	
Eogaudryceras (Eotetragonites) balmensis	Albian	0.98	
Eogunnarites unicus	Cenomanian	0.76	
Eupachydiscus haradai	Campanian	0.9	
Gabbioceras angulatum	Aptian	0.88	
Gabbioceras latericarinatum	Albian	0.93	
Gabbioceras michelianum	Albian	0.9	
Gaudryceras cf. denseplicatum	Turonian	1.34	
Gaudryceras cf. tenuiliratum	Campanian	1.4	
Gaudryceras denseplicatum	Turonian	1.52	
Gaudryceras stefaninii	Cenomanian	0.93	
Gaudryceras striatum	Campanian	1.26	
Gaudryceras tombetsense	Maastrichtian	1.42	
Hauericeras angustum	Campanian	0.7	
Hauericeras gardeni	Campanian	0.72	
Holcophylloceras guettardi	Aptian	0.81	
Holcophylloceras sp.	Aptian	0.81	
Hoploscaphites comprimus	Maastrichtian	0.66	
Hoploscaphites nebrascensis	Maastrichtian	0.68	
Hoploscaphites nicolletii	Maastrichtian	0.77	
Hoploscaphites spedeni	Maastrichtian	0.76	
Hypacanthohoplites sp.	Aptian	0.84	
Hypacanthohoplites subcornuenianus	Aptian	0.93	
Hypophylloceras hetonaiensis	Maastrichtian	0.9	
Hypophylloceras ramosum	Maastrichtian	1.02	
Hypophylloceras subramosum	Santonian	1.03	
Karsteniceras obatai	Barremian	0.75	
Kossmatella agassiziana	Albian	1.11	
Luppovia sp.	Aptian	0.7	
Mantelliceras japonicum	Cenomanian	0.89	
Marshallites compressus	Cenomanian	0.97	
Melchiorites sp.	Albian	0.69	
Menuites pusilus	Maastrichtian	0.87	
Menuites yezoensis	Maastrichtian	0.76	
Metaplacenticas subtilistriatum	Campanian	1.14	
Microdesmoceras tetragonum	Cenomanian	0.94	
Nolaniceras sp.	Aptian	0.79	
Parahoplites melchioris	Aptian	1.21	
Parajaubertella kawakitana	Cenomanian	1.12	
Phylloceras japonicum	Cenomanian	0.92	
Phyllopachyceras ezoense	Santonian	1.08	
Phyllopachyceras ezoense	Campanian	0.91	
Phyllopachyceras sp.	Aptian	0.76	
Protexanites minimus	Santonian	0.74	
Pseudohaploceras nipponicus	Aptian	0.79	
Pseudophyllites indra	Campanian	1.48	
Ptychoceras renngarteni	Aptian	0.85	
Ptychophylloceras ptychoicum	Berriasian	0.69	
Pusozia takahashii	Turonian	0.89	
Puzosia orientale	Turonian	0.83	
Puzosia pacifica	Turonian	0.84	
Puzosia yubarensis	Turonian	0.61	
Saghalinites teshioensis	Campanian	1.19	
Scaphites carlilensis	Turonian	0.6	
Scaphites corvensis	Turonian	0.67	
Scaphites depressus	Coniacian	0.76	
Scaphites larvaeformis	Turonian	0.59	
Scaphites nigricollensis	Turonian	0.68	
Scaphites planus	Turonian	0.84	
Scaphites preventitricosus	Turonian	0.65	
Scaphites pseudoaequalis	Coniacian	0.72	
Scaphites warreni	Turonian	0.66	
Scaphites whitfieldi	Turonian	0.65	
Scaphites yonekurai	Coniacian	0.87	
Simbirskites coronatiformis	Hauterivian	1.09	
Simbirskites discofalcatus	Hauterivian	1.17	
Simbirskites elatus	Hauterivian	1.06	
Simbirskites sp.	Hauterivian	1.26	
Simbirskites sp.	Hauterivian	0.98	
Simbirskites versicolor	Hauterivian	1.09	
Subprionocyclus bakeri	Turonian	0.75	
Subprionocyclus minimum	Turonian	0.74	
Subprionocyclus neptuni	Turonian	0.78	
Teshioites sp.	Campanian	0.92	
Tetragonites duvalianus	Albian	1.06	
Tetragonites glabrus	Turonian	1.08	
Tetragonites hulensis	Albian	1.04	
Tetragonites minimus	Turonian	0.98	
Tetragonites popetensis	Campanian	0.97	
Tetragonites popetensis	Campanian	1.08	
Tetragonites terminus	Maastrichtian	1.8	
Texanites kawasakii	Santonian	0.93	
Tragodesmoceroides subcostatus	Turonian	0.88	
Valdedorsella akuschaensis	Aptian	0.68	
Yezoites klamathensis	Coniacian	0.75	
Yezoites matsumotoi	Coniacian	0.8	
Yezoites puerculus	Coniacian	0.82	
Yokoyamaoceras ishikawai	Turonian	0.89	
Zelandites aff. inflatus	Cenomanian	1.22	
Zelandites kawanoii	Santonian	1.19	
Zelandites mihoensis	Coniacian	1	
Zelandites varuna	Maastrichtian	1.22	
Zuercherella falcistriata	Aptian	0.75	

Table 2 Size of conchs of Thecosomata with coiled conchs.

Data sources are indicated in the table.

Taxon	Age	Conch size (mm)	Source	
Limacinidae	Eocene	1.125	Lokho & Kumar (2008)	
Limacinidae	Eocene	1.05	Lokho & Kumar (2008)	
Limacinidae	Eocene	0.733333	Lokho & Kumar (2008)	
Limacinidae	Eocene	0.75	Lokho & Kumar (2008)	
Limacinidae	Eocene	0.32	Lokho & Kumar (2008)	
Limacinidae	Eocene	0.568182	Lokho & Kumar (2008)	
Limacinidae	Eocene	0.857143	Lokho & Kumar (2008)	
Altaspiratella elongatoidea	Eocene	1.416667	Janssen & Goedert (2016)	
Altaspiratella elongatoidea	Eocene	1.833333	Janssen & Goedert (2016)	
Altaspiratella elongatoidea	Eocene	1.233333	Janssen & Goedert (2016)	
Heliconoides mercinensis	Eocene	1.764706	Janssen & Goedert (2016)	
Limacina aegis	Eocene	1.4375	Janssen & Goedert (2016)	
Limacina aegis	Eocene	1.71875	Janssen & Goedert (2016)	
Limacina aegis	Eocene	1.03125	Janssen & Goedert (2016)	
Limacina aegis	Eocene	1.53125	Janssen & Goedert (2016)	
Limacina aegis	Eocene	1.84375	Janssen & Goedert (2016)	
Limacina aegis	Eocene	1.28125	Janssen & Goedert (2016)	
Limacina novacaesarea	Eocene	1.382353	Janssen & Goedert (2016)	
Limacina novacaesarea	Eocene	1.588235	Janssen & Goedert (2016)	
Limacina novacaesarea	Eocene	1.470588	Janssen & Goedert (2016)	
Heliconoides lillebaeltensis	Eocene	2.4	Janssen, Schnetler & Heilmann-Clausen (2007)	
Heliconoides lillebaeltensis	Eocene	2.5	Janssen, Schnetler & Heilmann-Clausen (2007)	
Heliconoides lillebaeltensis	Eocene	2.5	Janssen, Schnetler & Heilmann-Clausen (2007)	
Heliconoides lillebaeltensis	Eocene	2	Janssen, Schnetler & Heilmann-Clausen (2007)	
Heliconoides lillebaeltensis	Eocene	1.7	Janssen, Schnetler & Heilmann-Clausen (2007)	
Heliconoides lillebaeltensis	Eocene	1.8	Janssen, Schnetler & Heilmann-Clausen (2007)	
Heliconoides mercinensis	Eocene	2	Janssen, Schnetler & Heilmann-Clausen (2007)	
Heliconoides mercinensis	Eocene	2.6	Janssen, Schnetler & Heilmann-Clausen (2007)	
Limacina pygmaea	Eocene	1.4	Janssen, Schnetler & Heilmann-Clausen (2007)	
Heliconoides sp.	Campanian	1.56	Janssen & Goedert (2016)	
Heliconoides mercinensis	Pleistocene	2	Guess	

The role of r-strategy in ammonite and belemnite ecology

Depending on the proportional gonad size and whether or not ammonites were semelparous or iteroparous, it appears likely that adult females of the largest puzosiid ammonites such as Parapuzosia seppenradensis laid between 10,000,000 and 100,000,000 eggs and ammonoids about half the size still over 1,000,000 eggs. The simple calculation above highlights the likelihood that derived ammonites were extreme r-strategists (respectively fast life strategy), which produced vast amounts of offspring, likely contributing an important part of the plankton in size at the limit from micro- to macroplankton. R-strategy corresponded with high mortality and it is likely that hatchlings and juveniles of ammonites formed a major source of food in the marine realm.

Table 3 Size of conchs of Thecosomata with straight conchs.

Data sources are indicated in the table.

Family	Age	Conch size (mm)	Source	
Creseidae	Eocene	1.875	Lokho & Kumar (2008)	
Creseidae	Eocene	1.25	Lokho & Kumar (2008)	
Creseidae	Eocene	1.25	Lokho & Kumar (2008)	
Creseidae	Eocene	0.75	Lokho & Kumar (2008)	
Creseidae	Eocene	2.142857	Lokho & Kumar (2008)	
Creseidae	Eocene	0.416667	Lokho & Kumar (2008)	
Creseidae	Eocene	0.597826	Lokho & Kumar (2008)	
Creseidae	Eocene	0.597826	Lokho & Kumar (2008)	
Creseidae	Eocene	0.480769	Lokho & Kumar (2008)	
Creseidae	Eocene	0.477273	Lokho & Kumar (2008)	
Creseidae	Eocene	0.575	Lokho & Kumar (2008)	
Creseidae	Eocene	0.5	Lokho & Kumar (2008)	
Cliidae	Eocene	1.857143	Lokho & Kumar (2008)	
Cliidae	Eocene	1.145833	Lokho & Kumar (2008)	
Cliidae	Eocene	0.9375	Lokho & Kumar (2008)	
Cliidae	Eocene	1.5625	Lokho & Kumar (2008)	

Table 4 Estimates of numbers of eggs produced by various ammonoid taxa.

Genus	Age	Adult size (mm)	Fecundity	Source	
Parapuzosia	Cretaceous	2,000	10,000,000	This paper	
Pachydesmoceras	Cretaceous	1,000	3,000,000	This paper	
Sinzovia	Jurassic	50	200	Mironenko & Rogov (2016)	
Manticoceras	Devonian	400	200,000	De Baets et al. (2012)	
Erbenoceras	Devonian	150	500	De Baets et al. (2012)	
Mimosphinctes	Devonian	90	35	De Baets et al. (2012)	
Gyroceratites	Devonian	56	130	De Baets et al. (2012)	
Agoniatites	Devonian	300	4,500	De Baets et al. (2012)	

As far as belemnites are concerned, their global abundance had already decreased during the Late Cretaceous (prior to the main extinction event), freeing ecospace for, e.g., some other coleoids (Iba et al., 2011). Nevertheless, coleoids with conical phragmocones such as belemnites, diplobelids, Groenlandibelus or Naefia share a small initial chamber and likely small embryonic conchs (Bandel et al., 1984). Above all, there are a lot of similarities between belemnite and ammonite hatchlings such as their small initial chambers, their overall hatchling size, their supposed habitats including the planktonic mode of life and the r-strategy reproductive mode (Ward & Bandel, 1987; Arkhipkin & Laptikhovsky, 2012; Doguzhaeva et al., 2014). Accordingly, we assume that belemnite fecundity was also high, although much lower than those of the puzosiid ammonites because of the much lower size difference between adults and embryos (about 100–1,000 eggs per female).

Figure 2 Relationship between adult conch size and the estimated number of eggs.

Data are displayed in Table 4. The variation seen in smaller species likely roots in differences of the embryo size.

Which animals ate ammonites?

Evidence for successful and unsuccessful predation on medium to large-sized ammonites is not rare but identifying the actual predator is possible only in very few cases (Keupp, 2012; Hoffmann & Keupp, 2015). Additionally, most hard parts of ammonites (conch and lower jaw) were likely crushed by the predators and quickly dissolved in the digestive tract, making ammonites as fossilized stomach contents improbable, although a few cases have been reported where juvenile ammonoid remains are preserved in stomachs of Jurassic ammonites (Keupp & Schweigert, 2015; Klug & Lehmann, 2015). It is even more difficult to find evidence for predators that fed on hatchlings and neanic juveniles of ammonites (dm < 10 mm), which must have occurred in vast numbers in the world’s oceans of the Mesozoic. These early post-hatching developmental stages probably lived in the water column because their conchs already had functional phragmocones and they are often found in black shales, which were deposited under hypoxic to anoxic bottom water conditions and therefore, a strictly benthic mode of life was impossible (Shigeta, 1993; Nützel & Mapes, 2001; Mapes & Nützel, 2009). Thus, pelagic nektonic animals (including older growth stages of ammonites) are the likeliest candidates as predators feeding on these young ammonites: Fig. 3 depicts the fitting of juvenile planktonic ammonites and holoplanktonic gastropods with the mesh size of filter feeders of the corresponding time intervals. See Fig. 4 for the stratigraphic distribution of the groups discussed here.

Figure 3 Zooplankton size ranks and filter mesh spacing of planktivorous filter feeders.

Modified after Vinther et al. (2014), using data from Lokho & Kumar (2008), Friedman et al. (2010) and De Baets, Landman & Tanabe (2015). The horizontal coloured lines connect the size ranges of zooplanktonic prey organisms with the mesh size of large filter-feeding marine vertebrates. The black lines mark the range in prey size organisms filtered by the respective filter feeders. Note how the size range of hatchlings of ammonites and belemnites match that of Thecosomata and, on the filter feeder-side, that of the mesh-size of the filtering organ.

Figure 4 Occurrences, extinctions, originations and diversity changes in plankton and large planktotrophic suspension feeders from the Cretaceous to the Palaeogene (mass extinction marked by red bar).

Ammonite and belemnite diversity show the number of species. Data from Friedman et al. (2010), Bristow, Ellison & Wood (1980), Corse et al. (2013), Yacobucci (2015) and Jarman (2001). Note the relative timing of extinctions of cephalopods with small planktonic juveniles and filter-feeding bony fish and rediations of holoplanktonic gastropods, krill as well as filter-feeding chondrichthyans and whales.

For abundant and easy prey like juvenile ammonites, a broad range of predators can be hypothesized. Like plankton today, these masses of juvenile ammonites represent perfect food sources for medium-sized to large suspension feeders (invertebrates and vertebrates). From the Cretaceous, giant planktivorous bony fishes (pachycormids: Friedman et al., 2010) have been suggested to be nektonic suspension feeders, which might have fed on plankton comprising a wealth of juvenile ammonites. In the SOM of their paper, Friedman et al. (2010) show a fragment of the gill rakers; their filaments have a spacing of about 1 mm, which is suitable to filter out hatchlings and juvenile ammonites with conchs of one millimeter in diameter or slightly more (Fig. 3). This trophic relationship is further suggested by the extinction of this group synchronous with the demise of the Ammonoidea and Belemnitida but direct evidence is missing. Taking the direct fossil evidence from the Jurassic into account, it appears likely that ammonites also played a role as micropredators feeding on early juvenile ammonite offspring (Jäger & Fraaye, 1997; Klug & Lehmann, 2015; Keupp, 2012; Kruta et al., 2011; Keupp & Schweigert, 2015; Keupp et al., 2016).

The extreme differences in size (up to three orders in magnitude) between adults and juveniles in large ammonites indicate that the range of potential predators changed significantly throughout the life history of these cephalopods. As hatchlings and small juvenile planktonic forms, moderate-sized to large suspension feeders and small predators likely used them as a food source but for adult puzosiids and other large ammonites, only large predators such as mosasaurs, pliosaurs and large fishes can be considered, although the seeming direct evidence for such a trophic relationship is still under debate (Kauffman & Kesling, 1960; Tsujita & Westermann, 2001; Kauffman, 2004; Gale, Kennedy & Martill, 2017). Latest Cretaceous juvenile ammonites were possibly not the primary food source of ichthyosaurs since the latter became extinct already in the Cenomanian and are unknown from stomach contents to our knowledge. After the demise of ichthyosaurs, ammonites persisted to be diverse and abundant. In spite of a better link of their extinction with that of the belemnites in the North Pacific near the end of the Early Cretaceous (Iba et al., 2011), belemnite decline in the Tethys at the Cenomanian–Turonian boundary (Doyle, 1992; Christensen, 2002) and direct evidence for a trophic relationship between phragmocone-bearing coleoids and ichthyosaurs (Kear, Briggs & Donovan, 1995 and references therein), Acikkol (2015) suggested that a link between the severe reduction of belemnite diversity and ichthyosaur extinction is unlikely.

An additional point we want to raise is the role of adult ammonites as micropredators feeding on juvenile ammonites. It has been claimed by various authors that ammonites, particularly heteromorphs, lived in the water column (Cecca, 1997; Guex, 2006), i.e., with direct access to meso- and microplankton. As pointed out above, a microphagous diet has been shown for several ammonite species including heteromorphs (Jäger & Fraaye, 1997; Klug & Lehmann, 2015; Keupp, 2012; Kruta et al., 2011; Keupp & Schweigert, 2015; Keupp et al., 2016), thus making ammonites as predators feeding on juvenile ammonites in the Cretaceous likely.

Which groups filled the ecospace freed by the extinction of ammonite hatchlings and planktivorous actinopterygians?

Association of the extinctions of large marine reptiles, large planktivorous fish and those of ammonites and belemnites suggest trophic relationships between these groups; their extinction freed ecospace for both small zooplankton and various suspension feeders as well as predators. This association is followed by major changes in the planktonic realm such as the rise of holoplanktonic gastropods. Although a few Early Jurassic to Cretaceous heteropods are known (Bandel & Hemleben, 1995; Nützel, 2014; Teichert & Nützel, 2015; Janssen & Goedert, 2016; Nützel et al., 2016; Burridge et al., 2017; Janssen & Peijnenburg, 2017), the major expansion of heteropods and ‘pteropods’ falls into the Cenozoic (Lalli & Gilmer, 1989; Tracey, Todd & Erwin, 1993; Janssen & Goedert, 2016; Janssen & Peijnenburg, 2013).

In size and their coiled form, many fossil Limacinidae (Thecosomata, planktonic opisthobranch gastropods) resemble ammonites. Similarly, the conchs of fossil Creseidae morphologically and in size (at least roughly) correspond to hatchlings of belemnites, diplobelids and other phragmocone-bearing coleoids of the Cretaceous (Bandel et al., 1984; Lokho & Kumar, 2008). In addition to these morphologic similarities, these groups shared the planktonic habitat. According to Janssen & King (1988), Janssen & Peijnenburg (1988), ‘pteropods’ were already present at least as early as the latest Palaeocene (see also Janssen & Goedert, 2016). Janssen & Goedert (2016) even claimed a Cretaceous origin of the Thecosomata. A number of Eocene pteropod occurrences are known worldwide (Bristow, Ellison & Wood, 1980; King, 1981; Curry, 1982; Zorn, 1991; Hodgkinson, Garvie & Bé, 1992; Janssen, Schnetler & Heilmann-Clausen, 2007; Lokho & Kumar, 2008; Ando, Ujihara & Ichihara, 2009; Cahuzac & Janssen, 2010). An early Palaeogene origin is also supported by a combination of palaeontological and molecular clock data published by Corse et al. (2013). Remarkably, the latter authors compare the uncoiling of the conch of Thecosomata with the coiling of ammonites, but they did not discuss macroecological implications such as the ecological replacements suggested here. As far as abundance of these fossils is concerned, pteropods are much less frequent than subadult to adult ammonites and belemnites, while their hatchlings are similarly rare. This is probably due to the combination of their small body size as well as their thin and fragile aragonitic shells (Janssen & King, 1988), which did not provide a high fossilization potential. The great majority of these thin aragonitic shells was undoubtedly rapidly dissolved during early diagenesis and as a consequence not fossilized (Berner, 1977; Janssen & Peijnenburg, 2017). Nevertheless, the fact that numerous pteropods have been reported from the Eocene implies that they were abundant and widely distributed at least since that period of time.

Similarities in size, overall morphology, habitat, abundance as well as the timing of their respective extinction and origination suggest that hatchlings and small individuals of ammonites as well as belemnites were ecologically replaced, at least partially, by planktonic opisthobranchs (Thecosomata) and other holoplanktonic gastropods. In turn, the ecological installation of the Thecosomata together with other planktonic organisms contributed to the dietary basis for the evolution of new groups of large planktivorous suspension feeders (Lalli & Gilmer, 1989; Armstrong et al., 2005; Hunt et al., 2008). As suggested by Friedman et al. (2010), the Cretaceous ‘giant planktivorous bony fishes’ found an ecological replacement in both large suspension-feeding chondrichthyans and baleen whales. Several of these groups are known to take in important amounts of planktonic gastropods, although arthropods such as krill and other plankton plays important roles as well. Today, thecosomes may regionally contribute up to 50% of the zooplanktonic biomass and thus are ecologically important (Mackas & Galbraith, 2012). However, today’s Manta rays (Mobulidae) are known to feed predominantly on small Crustaceans, small fish and other plankton and the same holds true for several baleen whales (e.g. Sims & Merrett, 1997; Motta et al., 2010; De la Parra Venegas et al., 2011; Slater et al., 2017). Nevertheless, it is unknown what exactly these Palaeogene suspension feeders ate, but at least the filter mesh spacing of both planktivorous chondrichthyans and several baleen whales fits well with the size range of thecosomes (Fig. 3). Also, it is conceivable that early filter feeding species of these groups did not discriminate in their planktonic diet as much as more derived modern relatives. In any case, the niche of filterers catching prey of a size of a few millimetres was occupied by the chondrichthyan and marine mammal filter feeders mentioned above. The Thecosomata can thus be understood as an example for the new plankton that occurred after the end-Cretaceous mass extinction.

What caused this ecological replacement?

The fossil record comprises quite a few cases of ecological replacements (which are often called ‘biotic replacements’; e.g., Benton, 1987; Benton, 1991; Briggs, 1998). Some famous examples include replacements of brachiopods by bivalves (Gould & Calloway, 1980; Payne et al., 2014), dinosaurs by mammals (e.g., Sloan et al., 1986), hybodonts by modern sharks (Schaeffer, 1965), and ‘straight-necked’ turtles by ‘flexible-neck’ turtles (Rosenzweig & McCord, 1991; for a comprehensive list of ecological replacement, see Benton, 1987; Benton, 1991). These examples are also used to discuss possible mechanisms, which explain what drove the replacement and in turn, facilitated speciation or macroevolution of clades. Benton (1991) presented several models of ecological replacements, in which the role of competition, key adaptation and mass extinction was discussed as main cause of such replacements. He concluded that ‘competition’ and ‘key adaptation’ can rarely be the main cause of such replacements, although the role of mass extinction also needs to be further examined. By contrast, some other researchers have different views on mechanisms of ecological replacements. For instance, Rosenzweig & McCord (1991) wrote the following: “species from the new clade produce new species to replace already extinct species from the old clade. The key adaptation gives them a higher competitive speciation rate than old-clade sources of replacement”. They presented the example of the straight-necked turtles (Amphichelydia), which were replaced by cryptodiran turtles with the capability of neck retraction and termed this model of replacement ‘incumbent replacement’. In this mechanism, key adaptations play a major role. Also, Briggs (1998) argued that several cases of ecological replacement, which are cited as evidence for mass-extinction-induced replacements, took place over a long period of time, and thus, biological interaction played a primary role.

In the case of ammonoids and belemnites being ecologically replaced by holoplanktonic gastropods, it partially depends on the phylogenetic and temporal scale that is examined and on the fossil record. When focusing only on ammonoids and belemnites, no species remained after the Cretaceous to compete with Palaeogene holoplanktonic gastropods. By contrast, when including the whole clade, this looks different since cephalopods continued to exist and gastropods were also present long before this faunal change (Fig. 5; Sepkoski, 1981). What makes this more relevant is the end-Cretaceous diversity reduction among the Cephalopoda in a phase where the whole clade Gastropoda diversified (a longterm process that started already in the Triassic, which was largely a result of the expansion of Neogastropda). Naturally, both Cephalopoda and Gastropoda contain species with a broad range of different modes of life and thus, their differing diversity patterns cannot be explained by the ecological replacement of juvenile ammonites by holoplanktonic gastropods alone—especially although holoplanktonic gastropods are abundant and widely distributed, their diversity is not high. On the other hand, the removal of ammonites and belemnites from the planktonic realm offered certain gastropods the opportunity to flourish in this realm and to long for new opportunities including availability of food and space resources.

Figure 5 Diversity of Cephalopoda and Gastropoda through the Phanerozoic. Redrawn after Sepkoski (1981).

Note the diversity reduction in cephalopods at the end of the Cretaceous and the subsequent radiation of gastropods. The Cretaceous Palaeogene mass extinction is marked by the red line.

As far as key adaptations (e.g., Simpson, 1943; Benton, 1983; Benton, 1987; Rosenzweig & McCord, 1991; Briggs, 1998; Payne et al., 2014) are concerned, they were already present in the group of species assuming the ecological role of the previously incumbent species after their extinction (Rosenzweig & McCord, 1991). Although some researchers report the presence of holoplanktonic gastropods in the Cretaceous (Janssen & Goedert, 2016; Burridge et al., 2017), to date, there is no fossil evidence, which suggests that they acquired all key adaptations that ensured the success of radiation in the Cenozoic except the small, thin-shelled conch. Also, due to a lack of data, it is not possible yet to conclude whether or not a direct biotic interaction between the old (ammonite and belemnite hatchlings) and the new (holoplanktonic gastropods) clades occurred in the Cretaceous, although the Thecosomata originated already in the late Campanian (Janssen & Goedert, 2016; Burridge et al., 2017). At least, the fossil record suggests that this case of replacement was triggered by a mass extinction event. In Fig. 6, we show the distribution of sizes of embryonic conchs of ammonites compared to the earliest occurrences of Thecosomata in the fossil record demonstrating the size-overlap between the two groups. Moreover, the evolution of a holoplanktonic mode of life is particularly easy in gastropods with a planktotrophic veliger larva. Bandel et al. (1984) have convincingly shown that pteropods evolved through neotenic extension of larval life of a benthic ancestor. A similar model has been proposed by Teichert & Nützel (2015) for Jurassic heteropods, which are superabundant in the Early Jurassic Posidonia Shale. Here, repeated Early Jurassic Anoxia resulted in neotenic prolongation of larval life to a holoplanktonic adult life. Thus, the free-swimming veliger larvae may be seen as an preadaptation for a holoplanktonic mode of life in these gastropods and this way the timely overlap that is required for in the incumbency model applies.

Figure 6 Box plots of Cretaceous ammonitella diameters and Cretaceous to Palaeogene Thecosomata conch size.

Data (Table 1) from De Baets, Landman & Tanabe (2015), Laptikhovsky, Nikolaeva & Rogov (2017), Lokho & Kumar (2008), Janssen, Schnetler & Heilmann-Clausen (2007), Janssen, Sessa & Thomas (2016), Janssen & Goedert (2016), and Janssen & Peijnenburg (2017). The red line marks the Cretaceous-Palaeogene boundary. The numbers below the box plots give the sample size. Orange marks ammonitella data, while green marks gastropod data. Question marks indicate that fossils are present but the values indicate guesses: in the case of Valanginian ammonites, protoconchs are known and from that size, an ammonitella size close to 1 mm appears likely. In the case of Campanian and late Palaeocene Thecosomata, the only genus ranging back into the Campanian is Heliconoides (Janssen & Goedert, 2016; Janssen & Peijnenburg, 2017); Eocene specimens measure approximately 2 mm, hence this size estimate.

Nevertheless, the main radiation of the Thecosomata appears delayed until the Palaeocene-Eocene boundary judging from their fossil record (Janssen, Schnetler & Heilmann-Clausen, 2007; Janssen & Peijnenburg, 2017). This raises the question whether this delay in radiation is an artefact of the poor fossil record of the Thecosomata or whether the main radiation really occurred several million years after the Cretaceous. Considering that a species loss caused by a mass extinction event can require as long as 5–10 My for community recovery (Copper, 1989), it appears likely that the radiation of holoplanktonic gastropods began in the Palaeocene. It must be taken into account that adaptation to the new life style including optimization of resource exploitation needed time.

Although we cannot rule out key adaptations and biotic interaction as main causes of the ecological exchange of ammonites and belemnites by Thecosomata and other holoplantonic gastropods after the Cretaceous, it is probably safe to claim that the holoplanktonic gastropods opportunistically benefited from the ecospace vacated by the extinction of these two groups of cephalopods. Possibly, new finds of holoplanktonic gastropods from the early Palaeocene will push their radiation back in time closer to the extinction of ammonoids.

Conclusions

Large Late Cretaceous ammonites such as puzosiids reached sizes exceeding two meters in diameter. Their offspring has a conch size that is in stark contrast to the adult size; the embryonic conchs of many Cretaceous ammonites measure only about 1 mm in diameter at the time of hatching. This size relationship, conch geometry and anatomical proportions allow estimates of the number of offspring per female. Accordingly, the largest females might have laid between 10,000,000 (semelparity, small gonads) and 100,000,000 eggs (iteroparity, large gonads). Apart from this extreme example, the great abundance of ammonites, many of them of considerable size as adults, throughout the Mesozoic and the generally small size of their offspring implies that juvenile ammonites and belemnites played a fundamental role near the base of Mesozoic food webs, both as primary consumers and as food source for secondary consumers. We assume that Mesozoic oceans were full of small hatchlings and juveniles of ammonites and belemnites in the mm to cm size range. This part of the planktonic food chain vanished with the extinction of ammonites and belemnites but may have enabled the evolutionary and ecological rise of holoplanktonic gastropods, which occupy a similar size range, conch morphologies (coiled and straight) and trophic role. This underlines the importance of ecological differentiation between different ontogenetic stages. Gill raker filament spacing in huge pachycormids correspond in size to these juvenile ammonites and belemnites, potentially suggesting a trophic link in the light of the synchronous extinction at the end of the Cretaceous.

Here, we suggest that the ecospace formerly occupied by ammonite and belemnite juveniles was taken over during the post-Mesozoic rise of the Thecosomata (holoplanktonic heterobranchs) and other holoplanktonic gastropods. The ecological replacement of the pachycormids is a bit more difficult to explain. During the early Palaeogene, three important large planktivorous lineages of chondrichthyans occur; however, modern mobulids (Manta rays), for instance, are known to feed on planktonic crustaceans. Perhaps, stomach contents of exceptionally preserved specimens of Palaeogene planktivorous chondrichthyans will shed more light on the suspension feeders that, at least in their function as primary consumers, benefited from the thecosomes that ecologically replaced juvenile ammonites. In any case, it is likely that ammonite hatchlings had occupied a significant part of their particular plankton size class and were important prey for some filter feeders. This size class niche of small plankton was vacant after the Cretaceous and holoplanktonic gastropods are one important group filling this ecospace, probably in concert with other plankton.

Independent of the filter feeder-side, we conclude that in r-strategists, the young offspring can play a more important ecological role than their large adults. This case of ecological replacement underlines the significance of differences at which developmental stage the acme in ecological importance of an organism occurs.

Jakob Vinther (Bristol) generously provided an illustration, which was a valuable basis for one of our illustrations. We thank Christoph Steinweg, Lothar Schöllmann and Jan-Ole Kriegs (all Münster), Kishor Kumar (Uttarakhand, India), Kazushige Tanabe (Tokyo), and Klaus Bandel (Hamburg) for providing illustrations and allowing us to use them. We greatly appreciate the input of the reviewers (Margaret Yacobucci, Bowling Green State University and two anonymous reviewers) and the editor Kenneth De Baets (Erlangen).

Additional Information and Declarations

Competing Interests

Author Contributions

Animal Ethics

Data Availability

The authors declare there are no competing interests.

Amane Tajika and Christian Klug conceived and designed the experiments, performed the experiments, analyzed the data, wrote the paper, prepared figures and/or tables, reviewed drafts of the paper.

Alexander Nützel analyzed the data, wrote the paper, reviewed drafts of the paper.

The following information was supplied relating to ethical approvals (i.e., approving body and any reference numbers):

Only fossil specimens were included, so animal ethics approval was not required.

The following information was supplied regarding data availability:

Specimens are on display in the following museums: LWL-Museum für Naturkunde Münster, Germany, and Mikasa City Museum, Mikasa, Japan.

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
