# Peer review of "The old and the new plankton: ecological replacement of associations of mollusc plankton and giant filter feeders after the Cretaceous?"

_PeerJ, doi:10.7717/peerj.4219_

## Round 0.1 · original submission · Major Revisions

· Academic Editor

Major Revisions

I apologize for the delay, but I was waiting for a final review. I commend the authors for their interesting hypothesis and nicely illustrated manuscript. I would like to see this manuscript published, although rather as a review paper than a pure research paper. However, there are several crucial points which need to addressed before publication.

Title: To be honest, i would drop "incumbent“ or add a question mark as it is a bit misleading. Typical examples of incumbent replacement involves clades which have a substantial overlap in temporal range and niches range (Rosenzweig & McCord 1991). Even in such clades (Briggs 1998), incumbent replacement does not always apply (e.g., brachiopods versus bivalves Payne et al. 2014). There is too little evidence for incumbent replacement in your case (as you also discuss in the manuscript).

Incumbent replacement: you discuss that there is no overlap in their fossil record (which would exclude „incumbent replacement“ from the beginning). Although there is now some evidence for pteropods in the Cretaceous (Janssen & Goedert 2016; Janssen & Peijnenburg 2017), which is also supported by the latest divergence time estimates placing their origin firmly in the Cretaceous (Burridge et al. 2017). However, their major radiation seems to fall around the Paleogene-Eocene Thermal Maximum (Janssen, Sessa & Thomas 2016) – and not directly after the K/P-mass extinction (see comments by reviewer 1). Their subsequent radiation is worth discussing in greater detail in this context, their potential driver(s) and the potential relationship with filter feeders.

Other filter feeders and their prey: it would be necessary to cite relevant publications on the prey of extant filter feeders (several statements are made, but not backed up by references). Furthermore, even with the statements you make you would need to discuss in more detail why crustaceans or other zooplankton could not have been the prey of larger Mesozoic filter feeders to make your case (see also comments by reviewer 3).

Ammonoid size data: Your focus on ammonoids is justified considering the available data, but you need to do more with the available data. The ammonoid data is treated very coarsely, although more highly resolved data is available for the Cretaceous (De Baets, Landman & Tanabe 2015). To verify if there was an actual overlap in size at the end of the Cretaceous (or only developed later) you would need to make boxplots through time of ammonoids and compare them with the size range of the supposed oldest pteropods (or other heteropods or their larvae). This could be easily done by plotting sizes through the Cretaceous.

Heteromorph ammonoids as plankton feeders: Another aspects which is only briefly mentioned is that many (adult) ammonoids might have been „filter feeders“ or at least microphagous themselves (Keupp et al. 2016). What else would large heteromorphic ammonoids (e.g., Nipponites) would have fed on? Their extinction could speak for a major change in food web structure at the end of the Cretaceous.

Belemnites: you mention belemnite occassionally, but focus on ammonoids. The manuscript would benefit from discussing some additional details on belemnite hatchlings (Doguzhaeva et al. 2014; Laptikhovsky, Nikolaeva & Rogov) and the potential similarities in reproductive strategies with ammonoids (Ward & Bandel 1987; Arkhipkin & Laptikhovsky 2012).

Cited references: Several crucial references on pteropods as well as food items of jawed vertebrates are missing. Furthermore, several recent references relevant for your interpretation are not cited. As you present limited new data, this is more a review paper. It is therefore even more crucial that all relevant references are present to make your point.

Review paper: As there is very limited new data provided (you only calculate fecundities for Parapusozia) – this would fit better as review paper – to my knowledge nobody has reviewed the possible replacement of associations of mollusc plankton and giant filter feeders in this way. Please submit your revised version as a review paper [# Note from staff: PeerJ does accept review papers - so it is not a problem for you to reformulate in that way#].

In addition, to the comments by the reviewers, please also address the following points:

Line 19: replaced is maybe a bit much – occupy similar niches/habitats would be more appropriate
Line 43: flood-basalt-eruptions – this is still quite controversial, so maybe add „potentially“, it is clear there was volcanism and that i was involved somehow, but some of the reference you cite here are heavy on volcanism, but not the meteorite.
Line 140: you need to reference for the reproductive strategies of belemnites (Arkhipkin & Laptikhovsky 2012; Laptikhovsky, Nikolaeva & Rogov 2017).
Line 150; please cite some additional references (Jäger & Fraaye 1997; Ritterbush et al. 2014)
Line 157; Mapes and Nützel was published in 2009 as far as i know
Line 170-171; Might benefit from discussing „abberant“ (heteromorph) ammonoids with a planktonic mode of life which must have had rich food sources?
Line 178: Kaufmann & Kesling 1960; more recent references (Tsujita & Westermann 2001; Kauffman 2004; Gale, Kennedy & Martill 2017) are necessary to give this a „still under debate“ status.
Line 187-188: it would more prudent to turn it around – if ammonite hatchlings could have had a similar role as pteropods
Line 201-202: the latest reviews place the oldest know fossils in the Late Cretaceous (see above)
Line 207-208: macroecological implications – you do not discuss them either: rephrase?
Line 208-209: their preservation potential is probably related with the aragonitic composition of their shells (if at all) and small size; please cite references(Janssen & Peijnenburg 2017)
Line 223-224: you need to cite references stating how important they might be as a food source and for what organisms! Some estimate that they are not that important in general (Lalli & Gilmer 1989) – other highlight that they can be crucial particularly in higher latitudes as a food source (Armstrong et al. 2005).
Line 226: please cite a reference which discusses the diet of manta rays. In this context, it is important to note that „krill“ was probably already around in the Cretaceous too based on molecular clock estimates(Jarman 2001).
Line 233: it is a bit one-sided to just focus on Puzosia – why not take examples of both extremes or an adult size study of ammonoids versus hatchling to see the common or full range of fecundities around during this time. These data are available and would make this manuscript a more valuable research contribution.
Line 249: i would add potential before „trophic link“ as this is very speculative without direct data through stomach contents, etc. Some groups first appear after the K/P-extinction (e.g., Manta rays), but their current „prey“ items (Krill) were already around before.
Line 256: you need to provide reference when you discuss the diet of extant taxa!

Suggested References:

Arkhipkin AI, and Laptikhovsky VV. 2012. Impact of ocean acidification on plankton larvae as a cause of mass extinctions in ammonites and belemnites. Neues Jahrbuch für Geologie und Paläontologie - Abhandlungen 266:39-50.
Armstrong JL, Boldt JL, Cross AD, Moss JH, Davis ND, Myers KW, Walker RV, Beauchamp DA, and Haldorson LJ. 2005. Distribution, size, and interannual, seasonal and diel food habits of northern Gulf of Alaska juvenile pink salmon, Oncorhynchus gorbuscha. Deep Sea Research Part II: Topical Studies in Oceanography 52:247-265.
Briggs JC. 1998. Biotic Replacements: Extinction or Clade Interaction? Bioscience 48:389-395.
Burridge AK, Hörnlein C, Janssen AW, Hughes M, Bush SL, Marlétaz F, Gasca R, Pierrot-Bults AC, Michel E, Todd JA, Young JR, Osborn KJ, Menken SBJ, and Peijnenburg KTCA. 2017. Time-calibrated molecular phylogeny of pteropods. PLoS ONE 12:e0177325.
De Baets K, Landman NH, and Tanabe K. 2015. Ammonoid Embryonic Development. In: Klug C, Korn D, De Baets K, Kruta I, and Mapes RH, eds. Ammonoid Paleobiology: From anatomy to ecology Topics in Geobiology 43. Dordrecht: Springer, 113-205.
Doguzhaeva LA, Weis R, Delsate D, and Mariotti N. 2014. Embryonic shell structure of Early–Middle Jurassic belemnites, and its significance for belemnite expansion and diversification in the Jurassic. Lethaia 47:49-65.
Gale AS, Kennedy WJ, and Martill D. 2017. Mosasauroid predation on an ammonite – Pseudaspidoceras – from the Early Turonian of south-eastern Morocco. Acta Geologica Polonica. p 31.
Jäger M, and Fraaye R. 1997. The diet of the Early Toarcian ammonite Harpoceras falciferum. Palaeontology 40:557-574.
Janssen AW, and Goedert JL. 2016. Notes on the systematics, morphology and biostratigraphy of fossil holoplanktonic Mollusca, 24. First observation of a genuinely Late Mesozoic thecosomatous pteropod. Basteria 80:59-63.
Janssen AW, and Peijnenburg KT. 2017. An overview of the fossil record of Pteropoda (Mollusca, Gastropoda, Heterobranchia). Cainozoic Research 17:3-10.
Janssen AW, Sessa JA, and Thomas E. 2016. Pteropoda (Mollusca, Gastropoda, Thecosomata) from the Paleocene-Eocene Thermal Maximum (United States Atlantic Coastal Plain). Palaeontologia Electronica 19:1-26.
Jarman SN. 2001. The evolutionary history of krill inferred from nuclear large subunit rDNA sequence analysis. Biological Journal of the Linnean Society 73:199-212.
Kauffman EG. 2004. Mosasaur Predation on Upper Cretaceous Nautiloids and Ammonites from the United States Pacific Coast. Palaios 19:96-100.
Keupp H, Hoffmann R, Stevens K, and Albersdörfer R. 2016. Key innovations in Mesozoic ammonoids: the multicuspidate radula and the calcified aptychus. Palaeontology 59:775-791.
Lalli CM, and Gilmer RW. 1989. Pelagic snails: the biology of holoplanktonic gastropod mollusks: Stanford University Press.
Laptikhovsky V, Nikolaeva S, and Rogov M. 2017. Cephalopod embryonic shells as a tool to reconstruct reproductive strategies in extinct taxa. Biological Reviews of the Cambridge Philosophical Society:n/a-n/a.
Payne JL, Heim NA, Knope ML, and McClain CR. 2014. Metabolic dominance of bivalves predates brachiopod diversity decline by more than 150 million years. Proceedings of the Royal Society B: Biological Sciences 281.
Ritterbush KA, Hoffmann R, Lukeneder A, and De Baets K. 2014. Pelagic palaeoecology: the importance of recent constraints on ammonoid palaeobiology and life history. Journal of Zoology 292:229-241.
Rosenzweig ML, and McCord RD. 1991. Incumbent replacement: evidence for long-term evolutionary progress. Paleobiology 17:202-213.
Tsujita CJ, and Westermann GEG. 2001. Were limpets or mosasaurs responsible for the perforations in the ammonite Placenticeras? Palaeogeography, Palaeoclimatology, Palaeoecology 169:245-270.
Ward PD, and Bandel K. 1987. Life history strategies in fossil cephalopods. In: Boyle PR, ed. Cephalopod life cycles, Vol II. London: Academic Press, 329-350.

Reviewer 1 ·

Basic reporting

Literature references incomplete, important and relevant references missing (see pt. 3 and 4)

Experimental design

The manuscript is mainly speculation with relatively little new information, of which I cannot fully judge the importance (pt. 3 and 4).

Validity of the findings

The authors present several conclusions:
1. Ammonites changed ecological role during their life time, as indicated by their extreme increase in size during ontogeny. They do not seem to bring up evidence for this statement, but it seems to be quite obvious – few organisms could eat an ammonite with diameter >1 m, many could eat ammonitella of about 1 mm.
2. Large female ammonites produced very numerous offspring (r-selected), so juvenile ammonites likely were an important food source for planktivores.
3. After the end Cretaceous extinction, juvenile ammonites were replaced by holoplanktonic gastropods, large pachycormid fish by filter-feeding chondrichthyans and cetaceans (sharks and whales).
The authors present newly generated information only as a part of to pt. 2, i.e., estimating fecundity of large ammonites. I know very little about ammonites, and leave the evaluation of this (important) part of the manuscript to other reviewers or the editor. I am able to evaluate pt. 3, and in my opinion the authors do not show convincing evidence that holoplanktonic gastropods diversified shortly after the end Cretaceous extinction, thus can be said to replace juvenile ammonites – there is considerable evidence that they radiated after the start of the Eocene, at least 10 million years later. The authors do not cite references showing the radiation in the Eocene. In my opinion the paper should thus be rejected, or (if the information on the ammonite fecundity is by itself sufficiently novel) be rewritten without linking extinction of ammonites at the end of the Cretaceous and expansion of Thecosomata more than 10 million years later.

Additional comments

Remarks by line number:

42-44: ‘..organisms…were erased by the consequences of an impact in Mexico and flood-basalt eruptions in India..’, followed by references. This sentence reads as if there is agreement in the community that the main cause of the extinctions were the combined effects of impact and volcanic eruption, but nothing is further from the real state of affairs: I would argue that the references show that hardly anyone would say this, but most people would argue that either the impact (Schulte et al) or the flood basalt eruptions (Keller et al) were the almost exclusive cause of the extinctions. Replacing ‘and’ by ‘or would improve the sentence..
47: only planktonic foraminifera, benthic foraminifera do not suffer significant extinction (Culver, Alegret et al.).
48: the bivalve statement is too simplistic; their pattern of extinction can in my opinion not be described as being ‘partial or total disappearance’ – see e.g. Vilhena et al 2013 Nature Sci Repts.
51-52: Did the Puzosiidae live during the late Maastrichtian, i.e. did they become extinct at the end of the Cretaceous? How common were they (and potentially other large ammonites) in the ecosystem? (something about how common they were appears in line 125, but ‘quite common’ is not very informative).
90: please define ‘derived ammonites; - are these heteromorphs? Were Puzosiidae derived?
113: how big is the lectotype of P. seppenradensis?
119: anything know about semelparous-iteroparous lifestyles of large ammonites? Is it probable that such large organisms are semelparous??
136: why discuss belemnites if they were already in decline (from about 35 million years before the K/Pg extinction)?? Do we know anything about coleoids replacing them?
153: do we know enough to make such a sweeping statement ‘these early psot-hatching developmental stages’ – all ammonites? Only derived ones?
154-155: refer to buoyancy – Shigeta 1993, Lethaia?
162: reasonable argument, but did not larger ammonites eat smaller ones as well? (I see- said in lines 169-170) – so how important were these large planktivorous fish? Common enough to have major impact?
167: I do not see that coeval extinction of two groups confirms that one ate the other- especially not if they both became extinct at the end of the Cretaceous (ammonites did)- see also below (189)
178-185: I do not really see why the discussion whether extinction of belemnites is linked to that of ichthyosaurs is relevant to the main story in this manuscript, and also do not quite see how extinction of that predator would lead to extinction of its prey only in the Pacific.
182: CTB? Cretaceous Tertiary Boundary (rather than KPg)?
189: once again, I do not see that coeval extinction of groups during a major mass extinction (when many things went extinct) has anything suggestion to make as to their trophic relations.
191-206: In my opinion it just is not true according to most evidence that the associated extinction (during the KPg) coincided with (i.e., occurred at the same time) as the rise of holoplanktonic gastropods, specifically Thecosomata. If we look at some recent papers (as well as those cited in lines 202-204), then it becomes clear that the ‘rise of the holoplanktonic gastropods’ (if equated with the increase in diversity of thecosomata) considerably post dated the KPg (and the extinction of ammonites): we see a single taxon in the Campanian and the Paleocene, with increased diversity only by the earliest Eocene, i.e., more than 10 million years after the KPg extinction. I realize that the cited Corse et al. say that there was a diversifying event ‘just after the Cretaceous/Tertiary mass extinction’, but the paper that they cite for this statement in fact only describes occurrence of pteropods in the Eocene, which in my opinion cannot possibly be called ‘just after the Cretaceous Tertiary mass extinction’ – even in geology I do not call 10 myr later ‘just after’. I suggest that the authors look up recent articles by Janssen & Peijnenburg, 2017, An overview of the fossil record of Pteropoda (Mollusca, Gastropoda, Heterobranchia)Cenozoic Research 17(1), 3-10, and look at their nice compilation figure, Burridge et al., 2017, Time-calibrated molecular phylogeny of pteropods. PlosOne, 12(6): e0177325, who state ‘Using a fossil-calibrated phylogeny that sets the first occurrence of coiled euthecosomes at 79±66 mya, we estimate that uncoiled euthecosomes evolved 51±42 mya and that most extant uncoiled genera originated 40±15 mya.’ , and Janssen, A. W., Sessa, J. A., and Thomas E., 2016. Pteropoda (Mollusca, Gastropoda, Thecosomata) from the Paleocene-Eocene Thermal Maximum of the United States Atlantic Coastal Plain., Palaeontologia Electronica, Article 19 (3), 1-26. These all show diversification of holoplanktonic gastropods in the Eocene (as did the paper cited in Corse et al). I guess one could still compare the ecological role of holoplanktonic gastropods and compare to juvenile ammonites, but the authors argue that the latter replaced the former shortly after their extinction, and that is not supported by the literature cited above - see below.
213-214, 216-217: yes, quite a few pteropods have been reported from the Eocene, but that is NOT ‘just after the K/Pg extinction.
221-223: the baleen whales may have replaced giant Cretaceous planktivorous fish as to ther ecological role, but as shown in fig. 3, they only evolved in the earliest Oligocene (during the cooling of the Earth and establishment of Antarctic ice sheets, commonly (though not universally accepted) linked to diversification and increased abundance of diatoms; see e.g., Norris et al., 2013, Science, 341, 492; Marx & Uhen, 2010, Science 327, 993). So it seems to me not valid to link the origin/diversification of whales to extinction of large Cretaceous planktivorous fish more than 30 million years earlier.
228: there were no early Paleogene (=Paleocene-Eocene) whales. Please document early Paleogene chondrichthyans- provide reference (also 255: which 3 lineages? Provide reference).
252-253: there was no Paleocene expansion of Thecosomata (at least, we have no evidence for that), there was an Eocene (10 million years later) expansion.

·

Basic reporting

English is generally good; I have marked up some suggested word changes and typographical errors in the pdf of the manuscript.

References are very thorough and capture the context of the paper's topic well. They do need to be proofread more carefully, as there are inconsistencies in citation format, e.g., abbreviating vs. not abbreviating journal titles, capitalizing each word in book titles vs. not, including "Topics in Geobiology" and volume number vs. not.

The figures are beautifully constructed, but are not described or discussed in any detail in the text. They really need to be explained more both in the text and in the figure captions. For example, Fig. 2 does not provide units for the zooplankton plots. Are these numbers of species? Numbers of genera? A percentage? What is the reader supposed to notice from looking at these plots? In Fig. 3, what are the colored dots and black lines in the plot supposed to represent? What is the reader supposed to learn from this figure? How does this figure support the author's argument? It is worth the authors' time to describe the figures and their message within the text.

Experimental design

The topic of what role hatchling ammonoids and belemnoid played in Cretaceous planktonic food webs, and how they and their potential predators were replaced after the end-Cretaceous extinction, is an interesting one that has not been well-explored in the past.

The research questions are not presented very clearly in the introduction. The authors state that they consider two research questions (lines 54-59), but the first item is not phrased as a question at all. It would be helpful to rewrite this paragraph, making the questions as explicit as possible.

The methods used to estimate the fecundity of large Cretaceous ammonoids are clearly explained and make sense; these are the primary new contribution. Much of the rest of the paper involves synthesis of various observations derived from the published literature.

Validity of the findings

The authors primarily use valid existing published data and observations, plus reasonable new calculations of the likely gonad volume and egg production of the largest known Cretaceous ammonoid Parapuzosia seppenradense, to argue for the importance of ammonoid hatchlings in the zooplankton of Late Cretaceous oceans. They then discuss some likely predators on these hatchlings, as well as replacement taxa for both hatchlings and their predators after the end-Cretaceous extinction. While somewhat speculative, these discussions are supported by reference to observations in the existing literature.

One concern I have is that the belemnoids seem "tacked on" to the paper, with just brief discussion of their importance to Cretaceous planktonic food webs. It would be helpful if the authors could add a little more information and/or data about belemnoids to support their claims.

Additional comments

This paper makes an interesting and well-supported case for the importance of ammonoids and belemnoids in Cretaceous planktonic food webs. I do note a few key issues to address in a revision (in order of importance):

1) Reframe the paragraph that presents your research questions. Make the research questions actual questions, and ideally phrase them to go beyond the descriptive (e.g., "what groups might have...?") to test a hypothesis or prediction.

2) Make better use of the figures by describing them in the text, making clear what the reader should be learning from each one, and adding more information to the captions explaining figure components.

3) Amplify the discussion of belemnoids, which seems like an afterthought in this version of the manuscript.

4) Ensure references are consistent in their format.

Reviewer 3 ·

Basic reporting

Writing is generally clear but sometimes fuzzy in few sentences. Some typos also pervade the MS. This is minor but I recommend that the authors review the paper for wording/style.

L.23: fishES?

Introduction: maybe too short. Authors can add for instance some known examples of potential similar replacements in the fossil record.

l.32: mass extinction intervals: what does it means exactly? They are more short events than intervals. Maybe reword this sentence.

l.36-38: This needs a few references. Also, it may depend on if you are talking about taxonomic richness and/or functional diversity. See e.g. recent works of Foster et al. on the PT and TJ extinctions.

l.43: flood-basalt-eruptions => flood basalt eruptions

l.49: Ammonoids “evolved a great disparity”: I suggest to rephrase this sentence.

l. 50: “bizarre forms”: maybe refer directly to heteromorphs.

l.51: please rephrase the beginning of this sentence.

l. 53: “they also occurred worldwide and in great numbers”. I suggest to modify this sentence, e.g.: “they also abundantly occurred worldwide”

l.63: maybe add in the sentence if this is true for all ammonoids and periods (?).

l.66: a space is missing after “(iii)”.

l.85-89: if possible, could be nice to highlight abbreviations for measurements in italics.

l.90: “as demonstrated by”: I’m not very prone to use such terms in articles. Maybe “as shown by” (or equivalent) is more appropriate. Same apply l. 212 “undoubtedly” is likely too strong.

l.90: “derived ammonoids”: unclear, please add a definition and details.

l.113-117: Important point: I expect this is a mistake but please do not use L² to refer to a volume!! => L3.

l.118-120: iteroparity vs semelparity: what is the most realistic based on the known fossil and modern data (if any)?

l.121: “fecundity would further increase”: unclear for me, especially the link with growth of embryos after they were laid. Please add an explanation and potential references (even if these are for modern cephalopods).

l.124: => Cretaceous

l.134: “high fecundity corresponded with high mortality”: this may appear rather authoritative. You should explain why, is this always the case and add references on that topic.

l.137: are we sure that belemnites were in direct competition with other coleoids? With all coleoids?

l.136-140: In my opinion, such general sentence on belemnites should arrive earlier in the ms, e.g. in the same part authors present ammonoids and their first ontogenetic stages. It could also be nice to see a little bit more space devoted to belemnites in the introduction.

l.178-185: this sentence is really too long to be well understood.

l. 182 CTB: add definition of this acronym.

l.178-185, l.189-191 and at some other places in the ms: authors often link potential trophic relationships among organisms with associated extinctions among the same organisms. I don’t say this is not the case but this should be nuanced as this link is far to be evident and may result from environmental changes or other biotic interactions.

l.206-207: “compare the UNCOILING of the conch of Thecosomata with the COILING of ammonites”: in my opinion, this sentence sounds very strange. What are the potential “macroecological implications”?

l.212-213: may add references on that point.

l.213-215: a “few pteropods” “implies that they were abundant and widely distributed”… strange wording I think. Please clarify the sentence by making a better link with previous sentences.

l.219: “instalment”: what does it means?

l.233: late => Late?

Fig. 2: Maybe shift the gastropod data on the other side, near the ammonite and belemnite data, for clarity.

References: they are sometimes in chronological order, sometimes in alphabetical order. Maybe homogenize them if you have instructions from PeerJ.

Experimental design

No comments.

Validity of the findings

This is my first review for PeerJ and I am not sure how the following comments fit well here or should be placed in Comments for author.
This work mainly resembles a short review on some “ecological replacements” that are observed during the KT transition. The working hypothesis is interesting because results support that ammonites and belemnites eggs and embryos may have served as food resources during the Cretaceous and that their disappearance at the KT boundary led to extinction of their potential predators. Such an idea is not totally new. The hypothesis of their replacement by holoplanktonic gastropods in food webs is also interesting as well as the discussion about potential new predators feeding on these small-sized organisms. We have nevertheless only rare direct fossil evidences for such feeding; the final discussion/conclusion is therefore partly speculative, but presented as such. To slightly enlarge the discussion and maybe yield slightly more balanced scenario, authors may preliminary answer to some associated questions: for instance, can other potential candidates than holoplanktonic gastropods have also served as potential food resources for filter feeding predators. What about crustaceans at that time? Are they too small? Too rare? Also to complement the discussion: are similar potential replacements (not necessarily complete) between an ammonite subgroup and another marine clade that has been documented (or speculated) in the Paleozoic, Triassic or Jurassic? This might be a point to address. Just a (probably naive) thought: can we imagine that largest ammonites also fed on smallest ones, thus reducing the number of available preys for other groups leading to an “equilibrium” inside the ammonoid clade?

Additional comments

This short MS presents an interesting working hypothesis on ecological/functional successions during the KT transition. This is relevant for PeerJ. I suggest to slightly enlarge some discussion points and I have highlighted a few areas that may benefit from some clarifications or rewording. I recommend publication after minor revision.

---

## Round 0.2 · Minor Revisions

· Academic Editor

Minor Revisions

Thank you for implementing our suggestions. By compiling additional data and references to back up some crucial statements, your manuscript has even become more excellent, easy to follow and of general interest. I like the new title and figures. Your paper is as good as accepted, there are just some minor points I would like to address before publication. These are:

Line 147: “superprising” instead of “surprisising”

Line 203: “suggested” would be more appropriate than “corroborated” as this is highly indirect evidence. Maybe taxa went extinct together around extinction events without a necessary causal link, but I agree that in this case, it is at least consistent with your hypothesis.

Line 217-218: “since the latter became extinct already in the Cenomanian”: it would be cautious here. I would drop this as extinction cannot say much about this relationship without other indications. I would argue rather with the fact the size of ammonoid hatchlings suggest they were not the main food source.

Figure 2 - Could you please add the putative Cretaceous fossil Thecosomata and more recent divergence time estimates (Burridge et al. 2017) as you do discuss them in the text, but they are not (yet) in this figure. You could add them as a dot and stippled lines respectively. It would make the figure even nicer than it already is and more consistent with the text.

I hereby all take the opportunity to provide you with some remarks on your rebuttal (not necessarily relevant for this manuscript, but to consider for future manuscripts).

As for the difference between abundance and diversity. Yes, it is hard in the fossil record. Nevertheless, the fact that you can find pteropods abundantly around the PETM where acidification might have been going on (even easier to dissolve their shells), does suggest that pteropods reach great abundance and diversity at least since the Eocene and potentially not before.

As for the references, I agree that it is hard to cite all literature, but some important points were not backed up by references and/or data previously. As for the literature, I apologize as I might be aware of more literature (at least on this particular subject) than most reviewers. In your manuscript, cited literature is now done in an exceptional and appropriate way – thank you.

One more remark: You are quite harsh on review papers (e.g., Ritterbush et al. 2014), but sometimes they also present new hypotheses and appropriately cite the authors. The point of a review paper is not just saving citations, but at least in my perspective to review the state of the art of research at a certain time, compare different lines of evidence that original author(s) did not do, identify points of uncertainty and launch new hypotheses to test. And yes, they can also be used to back up certain claims without citing a lot of original articles which only presented some of these points. Some journals limit the number of references for (review) papers, which is the main thing I have an issue with, as you should be able to give all appropriate and particularly all crucial papers in your field credit.

---

## Round 0.3 · accepted · Accept

· Academic Editor

Accept

Thank you for making these final changes. Looking forward too seeing this manuscript published.